# A Uniform, Tessellated Architecture for Energy-Efficient Learning and Inference

Jerry Felix*, Steve Brunker*, Carol Hibbard*

*Brain-CA Technologies, Inc. *jfelix@brain-ca.com,sbrunker@brain-ca.com,chibbard@brain-ca.com*

*Abstract*—This paper presents a novel architecture *designed from first principles* to support real-time learning and inference through the interaction of energy-efficient components. Each element operates autonomously, storing knowledge, reacting to inputs, and forming predictions without centralized control, complex arithmetic, or external scheduling. Inspired by biological systems and implemented as a uniform grid of identical cells, the architecture enables scalable, distributed intelligence using wave-based communication, collision-driven memory, and geometry-based predictions. We describe the logical and physical layers, detail subsystem roles, and explain how small components, tessellated across space, can deliver sophisticated behavior, suggesting a shift in AI design.

## I. INTRODUCTION

As artificial intelligence systems grow in size and complexity, their energy and resource demands have escalated dramatically. State-of-the-art foundation models require billions of parameters and exaflops of training compute, limiting their deployment in edge devices, sensors, and other energy-constrained environments.

This paper presents a fundamentally different architecture—one that reconsiders the foundations of learning and inference. Rather than relying on floating-point operations, centralized control, or layered backpropagation, this architecture builds intelligence from the ground up. It is based on the interaction of simple, uniform processing elements that store, communicate, and adapt through spatial alignment and timing.

Designed from first principles, the system minimizes component complexity and avoids energy-intensive operations like matrix multiplication or gradient descent. While some local arithmetic may occur, learning relies on discrete events and rule-based adaptation. Global coordination is replaced by *emergent behavior* across a grid of autonomous cells.

Inspired by biological systems—where learning is incremental, contextual, and energy-efficient—this architecture supports scalable, distributed intelligence through purely local interactions. By starting with the smallest learnable unit, we enable real-time embedded AI—whether at the edge or in the data center—without the complexity or cost of conventional neural systems.

## II. DESIGN PHILOSOPHY: FROM FIRST PRINCIPLES

Evolution favors energy-efficient intelligence. Organisms that spend less energy to react correctly have a survival advantage [1]. Similarly, efficient artificial systems should learn not by brute force but by interaction. The system mimics early biological intelligence—classifying stimuli (e.g., edible vs. harmful) based on exposure, not pretraining. It builds predictions incrementally using timing and local correlation—not through backpropagation [2] but via direct experience.

### A. Historical Context

Progress in technology often stems from identifying elemental components—such as the transistor, NAND gate, and RISC instruction. These breakthroughs simplified complexity into scalable patterns. Our architecture takes a similar approach, beginning with the Estimator—a micro-component that models a single binary data stream. From this elemental component, larger networks are formed by linking Estimators through spatiotemporal observation, echoing the modular composability that fueled revolutions in digital computing [3].

## III. ARCHITECTURAL OVERVIEW

The system is implemented as a 2D hexagonal grid of identical cells. Each cell communicates with its six immediate neighbors via directed signals, enabling both radial (wave-based) and channelized (path-based) communication. The architecture supports emergent learning through spatial signal interactions and wave collisions.

### A. Physical Design

Each cell integrates minimal hardware:
- **Local memory**: Stores persistent state (e.g., Estimator models, fast-path connections), and transient event metadata (e.g., pulse directions and relative origin locations).
- **Logic circuits**: Executes simple comparisons, bit flips, and state updates autonomously.
- **Communication buffers**: Temporarily hold signals traveling between neighboring cells.

Despite its simplicity, each cell contributes fully to the learning process. The design targets sub-microwatt power and scales to billions of instances per chip.

### B. Logical Design

Each cell contributes simultaneously to three logical subsystems, implemented in parallel:
- **Communication Subsystem**: Encodes metadata in waves. Each signal carries directional and temporal information encoded at emission and updated along the path.
- **Memory Subsystem**: Detects collisions, captures correlations, and updates Estimators based on the timing and alignment of incoming waves.

- **Connection Subsystem**: Dynamically forms high-speed paths (fast tracks) between frequently interacting cells to enable instantaneous predictions.

## IV. Learning Through Wave Collisions

In this architecture, learning emerges from detecting correlations between binary input signals arriving at different locations across a uniform, tessellated substrate. Each input—whether from a camera, microphone, LIDAR, or other sensor—is encoded as a binary value and delivered to a designated pulse point: a specific cell on the grid assigned to that stream. These pulse points emit waves that propagate outward, advancing one cell per clock tick by communicating only with immediate neighbors.

### A. Wave-Based Input Propagation

This clocked expansion keeps signals active just long enough to detect both simultaneous and slightly offset events. The use of slow, neighbor-to-neighbor wave propagation serves a crucial function: it allows inputs that are temporally close—but not perfectly aligned—to intersect during their traversal [4]. This property enables the system to discover patterns that may be concurrent, causal, or sequential in nature.

When two wavefronts intersect, the receiving cell—known as a **collision cell**—detects the interaction and initiates a learning process.

### B. Collision Cells and Spatiotemporal Inference

The collision cell compares the incoming signals and records their temporal relationship and binary values, then updates local Estimators to accumulate these events and detect consistent patterns over time.

At the point of collision, the system records the essential metadata needed to characterize the interaction—but critically, it does so without relying on absolute addresses or centralized memory. Instead, origin information is inferred through *relative* physical signal properties. The direction from which a wavefront arrives encodes its bearing, and the height (or intensity) of the wave indicates how far it has traveled from its origin. With all waves starting at uniform amplitude, distance is inferred from the wave's current discrete level—decreasing in fixed steps with each clock tick. No floating-point math is required—just an integer representing steps from its origin. The binary value of the original signal is encoded in the wave's shape or polarity—where a "hill" might represent a binary 1, while a "valley" indicates a 0 [5].

In this system, information is conveyed not only through value but through the structure of the signal itself. The geometry of wave propagation—timing, direction, and position—becomes a form of metadata. Without explicitly encoding identifiers or timestamps, each wavefront implicitly carries origin context and arrival delay. Spatial separation and timing offset jointly encode the history of the event, enabling local cells to infer relationships without any form of address-based lookup. In this way, **geometry becomes logic**—and space and time become integral dimensions of learning [6].

This mechanism can be understood through the analogy of a still pond. Dropping a stone creates a series of positive wavefronts, while pulling a plunger might create negative ones. When two such waves collide on the surface, their shape, timing, and direction of arrival offer enough information to infer both their relative distance and origin. Similarly, in this system, the shape and direction of digital wavefronts allow each cell to reason about where the pulses came from and what they represented—without any global coordinate system. This departs fundamentally from Von Neumann architectures [7], which rely heavily on addressable memory and fixed identity.

Once a correlation between two events is observed repeatedly, the collision cell stores the relationship using its local Estimators. A pair of fast paths is then constructed: one linking the collision cell to the origin of the first signal, and another connecting the collision cell to the location of the second event. These paths enable rapid prediction during future occurrences. When the first event recurs, its signal can take a shortcut through the fast path, allowing the system to deliver a prediction at the appropriate location and clock tick with minimal delay.

### C. Sequenced Predictions and Anomaly Detection

Because the system supports timing-based collision detection, it can identify and encode **sequenced predictions**—cases where one input reliably follows another with a brief delay. This feature mirrors a critical function of biological intelligence [8]. For example, when walking, the brain predicts what the foot should feel as it touches the ground. If the sensation differs—say, due to ice or a missing step—an alert is triggered. In both cases, prediction is used to filter expected outcomes and focus attention on surprises.

This architecture achieves the same result: when a pattern is correctly predicted, no alert is triggered. But if an input deviates from expectation, the discrepancy becomes salient to the system. In this way, the system supports both **predictive focus** and **anomaly detection** through simple, local rules and time-sensitive wavefront interactions.

## V. The Estimator: Elemental Intelligence

### A. The Estimator as a Minimal Learning Unit

At the core of this architecture lies a simple but powerful component: the Estimator. Inspired by biological efficiency and designed from first principles, the Estimator is a *self-contained learning unit* that models the statistical tendencies of a binary data stream using minimal energy, logic, and time. It solves a core task of biological learning: determining whether one outcome occurs more often than another—and their relative frequency.

The Estimator learns quickly, using only a few bits to form a model. Once formed, the model is continuously updated as new observations arrive, and can immediately be used to make predictions without additional computation. This makes it ideal for energy-constrained environments, supporting real-time prediction without delay or central control.

## B. Observation, Update, and Convergence

Internally, the Estimator maintains a sequence of storage bits, each paired with a randomly assigned bit. When a new observation arrives, the Estimator compares its storage bits to their random partners to determine the appropriate update:

- **If all pairs match**, a new storage bit is appended, matching the observed value.
- **If a mismatch is found**, and the observation matches the mismatched random bit, the model is incrementally nudged toward the observed value, using a small, rule-based structural adjustment.
- **Otherwise**, the observation is disregarded, as it contributes no new information to the model.

This selective update process subtly shifts the model's ratio over time to gradually reflect the statistical pattern of the input stream. Over time, this behavior causes the Estimator to converge on the ratio of 1s to 0s in the stream, but without using counters, division, or complex operations. Only simple bitwise updates are needed [9]. The model self-regulates: as more bits are added, updates shrink naturally—a form of statistical convergence where early inputs shape the model most strongly.

This design reflects a key principle of biological learning: randomness is not noise to be eliminated, but rather, a critical tool for generalization. Biological memory systems rely on stochasticity to prevent overfitting, support adaptation, and introduce variability into behavioral responses [10]. Similarly, our architecture embeds randomness directly into the learning process at the most fundamental level—each storage bit is paired with a random bit from the outset.

This pairing serves multiple roles:

- Prevents overfitting early signals by introducing variability in how observations influence the model.
- Regulates model growth by using fair random selection to decide when to expand or adjust, ensuring that updates are based on representative observations—not just frequency.
- Enables probabilistic prediction by embedding randomness into the structure, allowing the system to reflect real-world uncertainty without external noise injection.

Conventional neural networks rely on randomness primarily to initialize weights—effectively starting from a wrong answer that is later corrected through backpropagation [11]. This system instead begins from a blank slate. Each observation incrementally molds the model, with the first few having the strongest influence.

**Randomness is not added post hoc; it is embedded in the structure, enabling learning through selective, fair adaptation.**

## C. Prediction Modes

The Estimator supports two prediction strategies, both optimized for constant-time operation and ultra-low power.

- **Best Guess**: The system returns the most likely value based on the current model. This is achieved without computation—the Estimator simply reads the first (highest-order) storage bit. The Estimator tends to store higher-probability values earlier in the sequence, reflecting statistical dominance over time. This allows for constant-time predictions using minimal energy that are ideal for classification or decision tasks.
- **Probabilistic Mode**: The Estimator generates predictions in proportion to the observed ratio, using the embedded random bits to guide sampling. Internally, the model stores the binary outcomes in an order that tends to reflect dominance: the first bit represents the most likely half, the second the next quarter, and so on—mirroring a binary probability tree. To generate a probabilistic prediction, the system reuses the mechanism employed during learning: it identifies the first storage bit that differs from its paired random bit. This bit is selected as the prediction.

This method is extremely efficient. It relies solely on sequential bit comparisons—no counters, no division, and no separate sampling logic. Reusing the learning circuitry minimizes hardware and enables probabilistic inference with the same ultra-low energy cost.

## VI. PREDICTION AND INFERENCE

Once a collision cell observes a correlation between two binary signals, it becomes a predictive node. Using its local Estimators, the cell infers the likely value of one signal based on the other. Its position reflects the timing offset between inputs: midway if simultaneous, or closer to the latter signal if delayed.

This model supports two forms of prediction:

- **Immediate prediction** applies when signals are correlated in the same cycle—enabling anomaly detection or substitution if one is obscured.
- **Offset prediction** arises when one signal consistently precedes another. The system can act on this prediction immediately or deliver it later, aligned to the expected timing.

Unlike traditional systems, no explicit queue is required. Delayed predictions are routed slowly—at half speed—toward the spatial midpoint between signal origins. This cell naturally becomes the queue's head, with neighboring cells forming the sequence. When the prediction arrives, it is delivered at the correct moment via fast paths.

*Space thus encodes time.* Prediction sequencing and delivery are handled entirely through geometry, enabling temporally accurate inference without centralized control—ideal for dynamic environments and structured, anticipatory behaviors.

## VII. SUBSYSTEM BEHAVIOR AND EMERGENCE

### Communication Subsystem

The Communication Subsystem is responsible for propagating information across the grid. Unlike conventional systems that transmit explicit metadata, this architecture leverages the geometry of the wave itself. Each wavefront implicitly carries directional and temporal information through its method of propagation: the direction from which it arrives indicates

source orientation, and its height or timing reflects distance from origin. This allows receiving cells to interpret incoming waves relative to their spatial position, enabling the system to infer origin, timing offset, and event polarity without the need for embedded headers or tags.

*Memory Subsystem*

The Memory Subsystem monitors wave interactions and updates the local Estimator with each collision to accumulate evidence of potential correlations. When a strong pattern emerges, it may initiate fast path construction. It also manages prediction queues by routing inferred outcomes through spatially defined timing paths.

*Connection Subsystem*

The Connection Subsystem constructs and refines high-speed signal paths between frequently associated cells. These fast paths are created when predictions prove reliable and are reinforced through repeated use. They serve to bypass slow wave-based propagation and enable low-latency inference. Over time, paths that are no longer used or whose predictions lose accuracy are pruned, allowing the system to reallocate resources toward more relevant associations.

*Emergence*

No cell is specialized or externally assigned a role. Instead, intelligence emerges from repeated execution of simple local rules across the grid. Each cell supports all subsystems—communicating, remembering, and connecting—based on local inputs. As interactions accumulate, the grid self-organizes into a distributed engine for memory and prediction, enabling learning, inference, and adaptation at scale.

## VIII. Scalability and Energy Efficiency

The architecture's simplicity enables high-density fabrication and replication. Each cell performs only lightweight operations (bit comparison, flipping, signal routing), making energy cost negligible. Unlike neural networks that require thousands of floating-point operations for inference [12], this system performs inference through signal routing and bit-based prediction. It can be implemented in standard CMOS, eliminating the need for exotic hardware.

## IX. Model Growth, Pruning, and Adaptation

Learning in this architecture operates on two interconnected levels: the Estimator, which models the characteristics of binary signal streams and their pairwise relationships, and the Connection Subsystem, which adapts the physical pathways used to deliver predictions.

At the local level, each Estimator incrementally improves its understanding of a binary stream or the relationship between two signals. As new observations are made, the Estimator either adjusts its internal model or extends it with additional bits, refining its prediction capabilities over time. The process is continuous, requiring no reset, controller, or training phase. The model simply grows as needed, adapting naturally to changing conditions and newly observed patterns.

At the network level, the **Connection Subsystem** builds and maintains fast paths between associated signals. These paths are reinforced through use and pruned when correlations weaken or become obsolete—ensuring the system adapts to current data without clinging to outdated relationships. This dynamic structure balances stability and flexibility, enabling efficient, real-time learning at scale.

## X. Wave Geometry and Signal Metadata

Information in this system travels not only in value, but in structure. Each wave carries **implicit metadata**—encoded in its geometry and propagation behavior:

- **Direction vector**, representing entry point and propagation path
- **Relative origin**, inferred from the direction and age of the wave
- **Signal age**, represented by wave height (i.e., number of ticks since emission)

Rather than storing this data explicitly, the system decodes it from the wave itself. The **geometry of the wave**—when and where it collides with others—determines the resulting behavior.

This method turns geometry into logic: spatial separation and timing offset jointly encode the history of events. Local cells decode this structure to infer relationships—eliminating the need for explicit addressing or timestamp fields. Thus, **space and time become functional dimensions of computation**—rather than storage burdens—a core departure from neural architectures.

## XI. Conclusion

We present an architectural breakthrough: a scalable prediction system built from tessellated sub-cell units, spatial timing, and wave-based memory. With *no complex arithmetic*, *no controller*, and minimal energy, the system offers a compelling alternative to conventional inference engines—one that mirrors the efficiency and adaptability of biological intelligence, without attempting to replicate its structure.

Its simplicity extends to hardware realization. Each sub-cell contains a minimal, self-contained logic and memory structure, implemented using approximately 2,000 transistors. These units tile into hexagonal cells that scale efficiently across large grids with low power consumption and high fault tolerance.

Applications include embedded learning in sensor networks, fast-response prediction in autonomous systems, and anomaly detection in microcontrollers. Future research will explore analog extensions, formal convergence guarantees, and integration with symbolic reasoning for hybrid architectures.

Whether deployed at the edge or in large-scale systems, this architecture shows that intelligence can emerge from simple, learnable units—prioritizing learning over complexity.

*This work signals a shift in AI—from simulating intelligence to implementing the function of learning.*

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

## POSTSCRIPT: LIMITATIONS AND ETHICAL CONSIDERATIONS

### Limitations

While this architecture demonstrates scalable, low-power prediction capabilities, it assumes a uniform 2D substrate and synchronized tick-based signaling. Applications outside these constraints may require adaptation. Future work will explore heterogeneity, irregular grids, and mixed-modality inputs.

### Societal and Ethical Implications

As with any prediction technology, misuse is possible—for example, in surveillance or unintended autonomous control. Mitigation includes transparency in system design and limiting deployment to ethically aligned domains. The architecture's low-power nature could benefit underserved regions by enabling local intelligence without cloud dependence.

