# OpenReview forum: "A Uniform, Tessellated Architecture for Energy-Efficient Learning and Inference"
_iscaconf.org/ISCA/2025/Workshop/MLArchSys — MLArchSys 2025 Poster_

### Official Review · Reviewer_G7Kk · 2025-05-17
**The idea is novel but lack more evaluation and clear illustration**

**Confidence:** 3
**Rating:** 5

**Detailed Feedback And Questions For Authors:**

1.	It would be better to have a figure to illustrate their design and how the system works
2.	It would be better to have a minimum motivation experiment to show why their design could work or why we need these design.
3.	A more detailed illustration of how this method can outperform prior works is preferred.

**Top Reasons To Accept The Paper:**

1. I think the idea is quite novel. The paper proposes a fundamentally new paradigm: learning by local wave-collision and geometry rather than backpropagation or matrix operations.
2. It has a potential to be Ultra-Low-Power. By relying solely on bitwise logic, local memory, and simple signal routing, the design targets sub-microwatt per cell operation. If realizable, this could revolutionize on-edge AI for battery- and energy-constrained environments .
3. Their Decentralized design may be very useful

**Top Reasons To Reject The Paper:**

1. Lack of detailed illustration. Although the paper provides a lot of introduction to the different parts they designed, it is still not clear how these parts are connected and why such a design is required or preferred. I would suggest an illustration figure to help reader understand.


2. No Quantitative Evaluation. Even though this is a short paper, I think at least some motivation experiments about how these design can outperform others or why these design are promising is required. At least some quantitative comparison is needed.

3. Insufficient Comparison to Prior Art. Although they cite adaptive cellular automata and neuromorphic work, the submission lacks any head-to-head discussion of how this approach fares against state-of-the-art spiking-neuron chips or other energy-efficient inference engines (e.g., IBM’s TrueNorth, Intel’s Loihi). I think it needs to explain why their design would outperform or what the advantages of their design are, compared to prior works.

4. Unrealistic System Assumptions. The architecture hinges on a perfectly uniform 2D substrate and synchronized tick-based wave propagation. Real hardware is plagued by process variation, clock skew, and asynchronous events; the paper offers no robustness analysis under such non-idealities.

---

### Official Review · Reviewer_ynUD · 2025-05-17
**The premise is somewhat interesting, but the paper would benefit from a more concrete demonstration of the proposed architecture's use and more detail on the processes of the model.**

**Confidence:** 3
**Rating:** 3

**Detailed Feedback And Questions For Authors:**

This paper would benefit from a more concrete demonstration of the proposed architecture's use and more detail on the processes of the model (eg. updates, predictions, predictive node creation and function).

A major claim of the paper is that this model is highly efficient compared to traditional neural networks including for inference. This claim could use more support. For example, the scale of this sort of model is unclear. The model grows as it sees more observations (the Estimators store more bits). How much storage or computation is required to perform simple predictive tasks? The proposed model relies more heavily on memorization than traditional neural networks, which encode information with a constrained memory budget.

What constitutes an "observation" how exactly do the updates work? More detail here would be helpful in understanding this model. The observations include a binary value, direction and amplitude. How are the direction and amplitudes encoded or included in these updates? The Estimators are described as having a sequence of storage bits and random bits. There are only four possible states for two bit pairs. Does each of these Estimator sequences only contain two two bit pairs? The paper states: "If a mismatch is found, and the observation matches the mismatched random bit, the model is incrementally nudged toward the observed value, using a small, rule- based structural adjustment." How does this process work with only binary values?

The paper states "The Estimator tends to store higher-probability values earlier in the sequence". How and when are the order of the storage bits updated?

How are predictions made? What is the input to make a prediction? The paper states "the Estimator simply reads the first (highest-order) storage bit" to make a prediction in the best guess mode. How would the output of such a model be interpreted or used. A concrete example for this would be helpful in understanding the utility of such a framework.

**Top Reasons To Accept The Paper:**

The premise is somewhat interesting.

**Top Reasons To Reject The Paper:**

The description of the model, its learning and predictive processes are vague and would benefit from more specifics, diagrams, and examples. The model's size and complexity is unclear, especially in comparison to eg. simple or binarized neural networks. No numerical or experimental evidence is provided to demonstrate the use or effectiveness of the proposed model.

---

### Official Review · Reviewer_QcPb · 2025-05-18
**Novel architectural and learning paradigms that warrant future empirical validation**

**Confidence:** 3
**Rating:** 6

**Detailed Feedback And Questions For Authors:**

Quality: The paper presents a intriguing and novel conceptual ideas that aim to fundamentally rethink architectures for intelligent systems. The proposed learning mechanisms are compelling and offer a direction towards more biologically plausible computation. However, the overall quality is currently limited by the lack of empirical validation as far as function or efficiency goes.

Clarity: The paper provides a high-level description of an architecture, making the core ideas relatively accessible. Illustrations and possibly examples of processes would help the reader understand better.

Originality: The works significantly deviates from conventional compute and machine learning paradigms, especially its emphasis on geometry, space, and time as dimensions of learning are an under-represented approaches.

Significance: The papers ideas are intriguing to read and touch upon interesting and under-explored concepts in computer architecture and machine learning. If build upon and proven successful this paper could potential be a mile stone in the transformation in computer science.

**Top Reasons To Accept The Paper:**

- The paper describes a novel and compelling architecture that diverges considerably from conventional compute and machine learning principles.
- The learning rules, which operate incrementally through timing and local correlation, are an intriguing development that pushes towards more biologically plausible learning, thereby addressing limitations of backpropagation.
- The presented concept of "geometry becoming logic—and space and time becoming integral dimensions of learning" is appealing and represents a currently under-explored frontier. Ideas that effectively leverage these domains could indeed unlock new forms of efficient learning and provide novel insights into intelligence.
- The principled application of randomness as a feature to echo real world uncertainty without external noise injection, to mitigate overfitting, and to regulate model growth are well thought out.
- The system's inherent compatibility with standard CMOS fabrication processes is a crucial strength, making it relevant for practical implementation and aligning it closely with contemporary technological progress.

**Top Reasons To Reject The Paper:**

- Need for Empirical Evidence to Support "Architectural Breakthrough": The paper asserts an "architectural breakthrough," yet the manuscript primarily outlines high-level conceptual ideas. To fully validate this claim, the work would benefit significantly from a proof-of-concept, ideally through simulations demonstrating the architecture's learning behavior or efficiency.
- Oversimplification of Biological Energy Expenditure: The foundational premise, "Organisms that spend less energy to react correctly have a survival advantage," might overlook critical biological nuances. For instance, some biological systems prioritize accuracy over minimal energy expenditure, especially in scenarios where an incorrect response carries a high survival cost (e.g., as discussed in https://www.biorxiv.org/content/10.1101/2024.09.25.614654v2). The paper's own illustration regarding "edible vs. harmful" reinforces this point, where a wrong decision could be catastrophic.
- Clarification Needed on "Exposure, Not Pretraining" Claim: The assertion "exposure, not pretraining" could be more clearly articulated. In current machine learning paradigms, pre-training is widely understood as a process involving extensive exposure to a broad range of data. Therefore, the distinction between these two concepts could be further elaborated, as they are not necessarily mutually exclusive.
- Contextualizing Competition in the Modern Landscape: The paper draws parallels to historical abstractions like the NAND gate or RISC architecture, which gained prominence in less competitive environments. However, the proposed architecture faces an established ecosystem of highly optimized, high-performance systems. For widespread adoption, the proposed method would need to demonstrate profound, foundational advantages rather than incremental improvements, given the decades of optimization invested in current production systems.
- Inconsistency in Analogical Levels of Abstraction: The example "For example, when walking, the brain predicts what the foot should feel as it touches the ground" draws a comparison between a sophisticated, high-level brain function and what appears to be a more primitive function within the described architecture. This comparison of differing levels of complexity (e.g., a complex brain-level prediction versus a potentially simple cell-level process) might have limited persuasiveness.
- Requirement for Evidence to Support Claims of Intelligence Emergence: The authors claim that their work "shows that intelligence can emerge from simple, learnable units—prioritizing learning over complexity." As the paper does not include experiments or illustrative examples of such simple interactions, this significant claim currently lacks concrete substantiation.

---

### Official Review · Reviewer_qeKW · 2025-05-18
**High-level architecture proposal**

**Confidence:** 4
**Rating:** 2

**Detailed Feedback And Questions For Authors:**

I think the proposal is very interesting. However, this submission is not a good fit for an academic publication. There are several design choices presented that seem to come from a decision making process, however this process is not available in the paper. It should present alternative solutions, the trade-offs and why the particular choice is a good fit to the problem. The paper only presents a proposed design. And as is, it is very hand-wavy and has no evaluations.

**Top Reasons To Accept The Paper:**

- Proposes a new compute paradigm and a possible architectural implementation for it
- Promises very simple hardware components and energy efficient operation

**Top Reasons To Reject The Paper:**

- There is no grounding for the proposed ideas. The paper claims a possible shift in AI without demonstrating any concrete application for the proposed architecture.
- It reads as a set of high-level ideas to explore without any evaluation. It lacks a connection to the problem and does not provide any path on how to arrive at the solution. It does not consider any alternatives.
- It does not provide any scientifically acceptable evaluation methodology.